# Performance of self-reported measures of alcohol use and of harmful drinking patterns against ethyl glucuronide hair testing among young Swiss men

**Katia Iglesias** [1]*, **Séverine Lannoy** [2,3], **Frank Sporkert** [4], **Jean-Bernard Daeppen** [5], **Gerhard Gmel** [5,6,7,8], **Stéphanie Baggio** [9,10]

1 School of Health Sciences (HEdS-FR), HES-SO University of Applied Sciences and Arts of Western Switzerland, Fribourg, Fribourg, Switzerland, 2 Department of Psychiatry and Behavioral Sciences, Stanford University School of Medicine, Stanford, California, United States of America, 3 Laboratory for Experimental Psychopathology, Psychological Sciences Research Institute, Université Catholique de Louvain, Louvain-la-Neuve, Brabant Wallon, Belgium, 4 Center of Legal Medicine, Forensic Toxicology and Chemistry Unit, Lausanne and Geneva Universities, Lausanne, Vaud, Switzerland, 5 Addiction Medicine, Department of Psychiatry, Lausanne University Hospital, Lausanne, Vaud, Switzerland, 6 Addiction Switzerland, Lausanne, Vaud, Switzerland, 7 Centre for Addiction and Mental Health, Toronto, Ontario, Canada, 8 University of the West of England, Bristol, United Kingdom, 9 Division of Prison Health, Geneva University Hospitals and University of Geneva, Thônex, Geneva, Switzerland, 10 Office of Corrections, Department of Justice and Home Affairs of the Canton of Zurich, Zurich, Switzerland

* Katia.Iglesias@hefr.ch

## Abstract

### Background

There is a need for empirical studies assessing the psychometric properties of self-reported alcohol use as measures of excessive chronic drinking (ECD) compared to those of objective measures, such as ethyl glucuronide (EtG).

### Objectives

To test the quality of self-reported measures of alcohol use and of risky single-occasion drinking (RSOD) to detect ECD assessed by EtG.

### Methods

A total of 227 samples of hair from young Swiss men were used for the determination of EtG. Self-reported measures of alcohol use (previous twelve-month and previous-week alcohol use) and RSOD were assessed. Using EtG (<30 pg/mg) as the gold standard of ECD assessment, the sensitivity and specificity were computed, and the AUROC were compared for alcohol use measures and RSOD. Logistic regressions were used to test the contribution of RSOD to the understanding of ECD after controlling for alcohol use.

### Results

A total of 23.3% of participants presented with ECD. Previous twelve-month alcohol use with a cut-off of >15 drinks per week (sensitivity = 75.5%, specificity = 78.7%) and weekly

**Data Availability Statement:** The minimal anonymized data set necessary to replicate our study findings is available in the Supporting

information files. The full data set will be available at https://forsbase.unil.ch/project/study-public-overview/16799/1892/ from March 2021.

**Funding:** Grant awarded to SB and KI: Swiss National Science Foundation, no. 10001C_173418/1 URL: www.snf.ch/ The funders had no role in study design, data collection and analysis, decision to publish, or preparation of the manuscript.

**Competing interests:** The authors have declared that no competing interests exist.

RSOD (sensitivity = 75.5%, specificity = 70.1%) yielded acceptable psychometric properties. No cut-off for previous-week alcohol use gave acceptable results. In the multivariate logistic regression, after controlling for the previous twelve months of alcohol use, RSOD was still significantly associated with EtG (p = .016).

## Conclusion

Self-reported measures of the previous twelve months of alcohol use and RSOD were acceptable measures of ECD for population-based screening. Self-reported RSOD appeared to be an interesting screening measure, in addition to the previous twelve months of alcohol use, to understand ECD among young people.

## Introduction

Excessive alcohol consumption constitutes a major health issue worldwide [1, 2], accounting for approximately 3 million deaths (5.3% of all deaths) in 2016, of which 2.3 million were among men [3]. Excessive alcohol consumption mainly affects young people (20 to 39 years old), with mortality increasing up to 25% [1]. Furthermore, alcohol use is the third leading risk factor for poor health [4], with a strong economic impact ($250 billion cost for the USA with 2/5 total cost paid by the government [5]) and substantial societal consequences (e.g., suicides, violence, road traffic crashes, crime, victimization [4, 6]). It is worth noting that the highest levels of alcohol consumption per capita are observed in countries of the World Health Organization (WHO) European Region, with a 50% higher consumption (9.8 liters per capita in Europe versus 6.4 in the world population in 2016) [3], even if this consumption level has decreased since 2005 [3]. In Switzerland, even if epidemiological data show a decrease in the total alcohol consumption among young people [7], as in the rest of the European region [3], young people's alcohol use remains an important health issue. In addition, drinking substantial quantities of alcohol on a single occasion, the so-called risky single-occasion drinking (RSOD, drinking more than six units of alcohol at a single occasion for men), is very popular among young people (in Switzerland [8] and in the world [3]). More than 40% of young Swiss men report at least one monthly RSOD event during the last twelve months, a harmful drinking pattern likely to be maintained over a long period [9]. Therefore, there is a need for good indicators at the population level to monitor alcohol consumption, both among the whole population and among specific at-risk groups, such as young men. Indeed, self-reported measures of alcohol consumption have been largely criticized in the literature [e.g., 10], but it is worth noting that these measures provide critical information about alcohol consumption in large samples. For this purpose, investigating the reliability of self-reported measures using ethanol biomarkers would strongly reinforce the usefulness of self-reported measures. In particular, this information would help to develop adequate prevention policies, inform policymakers and minimize health, economic and societal burdens.

Ethyl glucuronide (EtG) is a direct ethanol biomarker used for abstinence control (urine) and assessment of risky and harmful drinking (hair). EtG is used in several settings, such as driver's license screening, workplace drug testing, child custody, liver transplantation, and as a condition of probation [1, 2]. Specifically, EtG in hair (hEtG) is a biomarker of long-term alcohol use due to its long detection window (depending on the length of hair), allowing for a retrospective evaluation of alcohol use based on an average anagen hair growth of 1.1 ± 0.2 cm per month [11]. In recent years, several systematic reviews or meta-analyses have documented

the reliability and validity of hEtG as an indicator of chronic excessive alcohol use defined as an average daily ethanol intake of 60 g [1, 2, 12, 13]. The last review on hEtG performed by Biondi and her colleagues [1] concluded that it is the most reliable and convenient biomarker to monitor alcohol use. hEtG has a high sensitivity and specificity with a cut-off of 30 pg/mg in the proximal 3-cm hair segment to detect alcohol use (i.e., >60 g/day). Importantly, these results are not influenced by age, sex, natural hair color, or the presence of liver disease [1]. Furthermore, even if there was some within- and between-laboratory variability, the impact was negligible, and the systematic review concluded that hair samples longer than 3 cm were acceptable [1]. A cut-off value for heavy drinking (>60 g of ethanol /day) of 30 pg/mg is proposed in several studies [e.g., 14]. This cut-off to define chronic excessive alcohol use using hEtG was established by the Society of Hair Testing (SoHT) based on consensus [15, 16] and was recently confirmed among young people [17].

Unfortunately, even if hEtG can be considered a reliable biomarker of excessive chronic drinking based on a threshold of 30 pg/mg [1, 12, 13], hEtG analyses are costly and time consuming and therefore not suitable for population-based assessments. Consequently, there is a need for well-validated self-reported measures of alcohol excessive chronic drinking compared to objective measures. To our knowledge, studies investigating the psychometric properties of self-reported alcohol use measures and of harmful drinking patterns against a gold-standard biomarker are not available. Therefore, we need empirical studies to determine how self-reported alcohol use and harmful drinking pattern measures converge with objective measures of excessive chronic drinking (i.e., hEtG).

This study aims to provide empirical evidence of the performance of self-reported alcohol use measures compared to objective biomarkers (i.e., hEtG). First, we would like to offer reliable measures of alcohol consumption and thus evaluate the usefulness of previous twelve-month alcohol use measures and previous-week alcohol use measures. Second, we would like to support the significance of RSOD measures and further explore its contribution to the understanding of excessive chronic drinking.

## Materials and methods

### Biological specimens

Data were collected among young Swiss men from the ongoing Cohort Study on Substance Use and Risk Factors (C-SURF) [18, 19]. The C-SURF cohort is composed of young Swiss men initially enrolled in army recruitment centers in 2010 (mean age at that time of enrollment, 20 years old). The military recruitment is mandatory in Switzerland. All males around age 20 are evaluated to determine their eligibility for military service, civil service, or no service without any pre-selection for this conscription. [19]. A subsample of this cohort was invited to participate in a new study (Screening for Alcohol Dependence among Young Swiss Men; SADYSM). SADYSM's participants were all alcohol users during the previous twelve months. The participants were selected using a random stratified sampling strategy based on the Alcohol Use Disorder Identification Test (AUDIT) score [20] to ensure the inclusion of sufficient participants with and without an alcohol use disorder [21]. The response rate of this study was 71%. For further details on the study design and sample selection, see the published protocol [22] and principal paper [23]. This study was approved by the Ethics Committee of the Canton of Vaud (no. 2017–00776).

Among the 233 young men recruited for the study, 227 agreed to give a hair sample for ethyl glucuronide determination. The length of the hair strands varied from 0.5 to 25 cm (mean: 4.2 cm, median: 3.0 cm). The assessments also included self-reported measures of

alcohol use (see measures section). Data were collected at the Lausanne University Hospital (Switzerland) from October 2017 to June 2018.

The patient information and informed consent forms complied with the guidelines of the Declaration of Helsinki (World Medical Organization 1996). All participants were informed in writing and orally about the aims of the study, why they were selected, what their involvement meant in terms of data collection, the risks/benefits of taking part and who the contact persons were for the study. They were also informed that they had the right to withdraw from the study at any time. Safeguards to ensure participant confidentiality were explained, and written informed consent was obtained from each participant prior to data collection.

## Analysis of EtG

Hair samples were analyzed for EtG using an in-house validated method based on micropulverized extraction [24] following the SoHT recommendations for sample preparation [15]. Hair was first washed with water and then acetone (5 min each). Between 5 and 40 mg dried hair (depending on the maximum available amount of hair) was cut in segments smaller than 2 cm and weighed into a 2 ml PP screw-cap microtube (Sarstedt, Germany). Two stainless steel balls (5 mm), 1 ml of water, and 5 ng EtG-d5 as an internal standard (5 μl of ethyl glucuronide-d5 at 1 μg/ml) were added to the hair. The closed tube was agitated for 30 min at 30 Hz in a Retsch 400 MM ball mill (Haan, Germany). After centrifugation, the extract was purified by SPE on Evolute$^®$ Express AX cartridges (6 ml, 150 mg, Biotage, Uppsala, Sweden). Washing was performed with 2 ml 5% aqueous ammonium hydroxide and 2 ml methanol, followed by elution with 2 ml methanol containing 2% formic acid. The eluate was evaporated at 30°C until the hair product was dry, and the product was then added to 50 μl of water containing 0.1% formic acid. Ten μl were injected into a Shimadzu 20 Prominence UFLC. Chromatographic separation was achieved on a Merck (Darmstadt, Germany) Chromolith$^®$ RP-C18e column (150 x 3 mm). Mass spectrometric detection was carried out on a Sciex 6500+ Qtrap operated in negative ESI mode. The transitions m/z 220.9–112.8, 220.9–84.9, and 220.9–74.8 were recorded for EtG and m/z 225.5–112.8, 225.9–84.9, and 225.9–74.8 for EtG-d5. Data treatment was performed using MultiQuant 3.0 software from Sciex (Toronto, Canada).

The calibration model was linear (weighting 1/x) with r > 0.99 for an EtG concentration range of 4–400 pg/mg. The limit of detection (LOD) and the limit of quantification (LOQ) were 2.0 pg/mg and 4.0 pg/mg, respectively. Coefficients of variation for two internal quality controls using commercially available authentic hair samples were ±19% at a reference value of 41 pg/mg (n = 50) and ±17% at a reference value of 23 pg/mg (n = 36).

## Measures

**Excessive chronic drinking.** Excessive chronic drinking was based on the threshold of 30 pg/mg with hEtG < 30 being non-excessive chronic drinking and hEtG ≥ 30 being excessive chronic drinking.

**Twelve-month alcohol use.** The previous twelve months of alcohol use were assessed with an extended quantity-frequency questionnaire providing separate information for weekdays and weekends over a period of time [25]. The number of drinks (corresponding to 10 g pure ethanol) per week was determined by multiplying the average frequency of alcohol consumption and the quantity of alcohol consumed.

**Previous-week alcohol use.** Alcohol use in the previous week was assessed by a past-week diary of the number of drinks consumed in the previous seven days for each day of the week, and the different kinds of alcohol were reported [25]. The total number of alcoholic drinks consumed over the week was computed.

**Risky single-occasion drinking.** RSOD was measured with the following question: "On the same occasion, how often do you drink six standard drinks or more?" on an ordinal scale: 'never', 'less than monthly', 'monthly', 'weekly' and 'daily'.

**Alcohol use disorder identification test.** The AUDIT is a ten-item questionnaire to screen for alcohol use disorders [20] and is also used as a screening tool for excessive chronic drinking [26]. The AUDIT includes questions on dependence (3 items), on specific consequences of harmful alcohol use (4 items) and on hazardous alcohol use (3 items). The score ranges between 0 and 40.

**AUDIT-C: The Alcohol Use Disorders Identification Test-Consumption.** The AUDIT-C includes the three first questions of the AUDIT, which focus on alcohol consumption (frequency, quantity, and excessive drinking episodes) [27, 28]. The score range between 0 and 12.

**Perceived family income.** Data were collected in the C-SURF cohort in 2010–2011 and merged with the SADYSM data. Participants were asked to report their perception of their family income: "How well off is your family compared to other families in your country?". Participants were grouped in two categories: in the average or below the average, *vs.* above the average.

## Statistical analysis

First, descriptive statistics (frequencies and percentages for categorical data; means, standard deviations (SD), median, and interquartile range (25–75%) for quantitative data) were computed for the whole sample and for the hEtG $< 30$ and for hEtG $\geq 30$ groups. We also computed the Spearman correlation between hEtG (continuous measure) and self-reported measures. Second, to assess the psychometric properties of self-reported measures of alcohol use and of harmful drinking patterns to assess excessive chronic drinking, sensitivity and specificity were calculated for the optimal cut-off values selected from the receiver operating characteristic (ROC) curves. The ROC curves of self-reported measures were compared using the areas under the ROC curves (AUROCs) to assess the best self-reported measure of excessive chronic drinking. The AUDIT and the AUDIT-C tools were also tested as they are widely used as self-reported screening tools. Third, we performed univariate and multivariate logistic regressions to explore (1) the predictive power of alcohol use measures and of harmful drinking patterns on excessive chronic drinking and (2) the contribution of RSOD to the understanding of excessive chronic drinking after controlling for alcohol use. We reported odds ratios (ORs) and confidence intervals (CIs), and as a measure of effect size, we computed the pseudo $R^2$ of McFadden (percentage of deviance of the model explained). Finally, as the SoHT [15] recommended hair length between 3 and 6 cm, a sensitivity analysis was run on a subsample of 129 participants with hair length between 3 and 6 cm. The overall results remained the same in this subsample, only two exceptions were noted in the comparisons between self-reported measures (i.e., between RSOD and AUDIT-C AUROC and AUDIT-C and Twelve-month alcohol use AUROC; see Table 2). Therefore, only the results for the whole sample are presented in the paper and the unique difference was mentioned in the results. The analyses of the 129 participants are available in the supplementary material (S1 Fig, S1–S3 Tables). Statistical analyses were performed using STATA version 14 (STATA Corporation, College Station, TX, USA).

## Results

### Descriptive statistics

Participants were, on average, 27.01 ± 1.45 years old. From the first C-SURF questionnaire (2010–2011), 41.0% of our sample have a perceived family income above the average, 48.9%

have a perceived family income in the average or below (10.1% missing data). For non-excessive chronic drinkers (hEtG < 30), 35.6% have a perceived family income above the average, 53.9% have a perceived family income in the average or below (with 10.9% missing data). For excessive chronic drinkers (hEtG ≥ 30), 58.5% have a perceived family income above the average, 34.0% have a perceived family income in the average or below (with 7.5% missing data).

Descriptive statistics of self-reported alcohol use measures and hEtG are reported in Table 1 for the whole sample and by levels of hEtG (<30 and ≥30). Correlations between hEtG and self-reported measures ranged between 0.47 and 0.60 (r = 0.60 for twelve-month alcohol use, r = 0.56 for RSOD (from daily to never) and r = 0.47 for previous-week alcohol use).

## ROC analysis

Twelve-month alcohol use had the best sensitivity and specificity to assess excessive chronic drinking with a cut-off of 15 drinks per week (75.5% and 78.7%, respectively), which were significantly better than the findings regarding previous-week alcohol use (p = 0.006), AUDIT-C (p = 0.024), and AUDIT (p = 0.314), but not significantly better than the findings regarding RSOD (p = 0.071) when AUROCs were compared. For the hair segment between 3–6 cm, RSOD was significantly better than AUDIT-C (p = 0.003) when AUROCs were compared. For further details, see Table 2 and Fig 1. The results were similar when taking into account for "perceived family income in or below the average" and "perceived family income above the average".

## Regression analysis

Finally, each self-reported measure of alcohol use (RSOD, Twelve-month alcohol use, and Previous-week alcohol use) was positively associated with hEtG (odds ratio between 3.8 and 10.7), with a pseudo $R^2$ between 7% and 20%. Combining RSOD, twelve-month alcohol use, and previous-week alcohol use, the best predictive model was obtained by combining RSOD and twelve-month alcohol use, with a pseudo $R^2$ of 22% (see Table 3).

**Table 1. Self-reported measures of alcohol consumption and hEtG for the whole sample and by level of hEtG.**

| | | Whole (n = 227, 100%) | | hEtG<30 (n = 174, 76.7%) | | hEtG≥30 (n = 53, 23.3%) | |
|---|---|---|---|---|---|---|---|
| | [min—max] | [1]mean +/- sd or % (n) | p50 [p25; p75] | mean +/- sd or % (n) | p50 [p25; p75] | mean +/- sd or % (n) | p50 [p25; p75] |
| RSOD | | | | | | | |
| Daily | | 3.1% (7) | | 1.1% (2) | | 9.4% (5) | |
| Weekly | | 34.4% (85) | | 28.7% (50) | | 66% (35) | |
| Monthly | | 28.2% (64) | | 31.0% (54) | | 18.9% (10) | |
| Less than monthly | | 24.2% (55) | | 29.9% (52) | | 5.7% (3) | |
| Never | | 7.1% (16) | | 9.2% (16) | | 0.0% (0) | |
| Twelve-month alcohol use | [0.4–91.0] | 13.8 +/- 13.6 | 11.5 [4.7; 18.0] | 10.9 +/- 11.1 | 9.1 [3.5; 14.5] | 23.3 +/- 16.5 | 18.5 [15.0; 24.0] |
| Previous-week alcohol use | [0–153] | 19.0 +/- 17.0 | 16.0 [7.0; 28.0] | 16.5 +/- 14.3 | 13.5 [5; 23] | 27.4 +/- 21.7 | 26.0 [16.0; 34.0] |
| AUDIT-C | [1–12] | 6.4 +/- 2.2 | 7.0 [5.0; 8.0] | 6.0 +/- 2.2 | 6.0 [4.0; 8.0] | 7.8 +/- 1.4 | 8.0 [7.0; 9.0] |
| AUDIT | [1–31] | 12.9 +/- 6.2 | 14.0 [8.0; 16.0] | 11.7 +/- 5.9 | 13.0 [6.0; 15.0] | 17.0 +/- 5.7 | 16.0 [14.0; 19.0] |
| hEtG | [0–691] | 24.0 +/- 53.8 | 10.0 [2.8; 28] | 8.4 +/- 7.8 | 5.2 [2.8; 13.0] | 75.3 +/- 94.1 | 53.0 [37.0; 70.0] |

[1]: mean +/- sd for quantitative variables and % (n) for categorical variables;

sd: standard deviation; p50: median; p25: first quartile; p75: third quartile; RSOD: Risky single-occasion drinking; AUDIT-C: The Alcohol Use Disorders Identification Test-Consumption; AUDIT: Alcohol Use Disorder Identification Test; hEtG: ethyl glucuronide in hair.

**Table 2. The diagnostic performance of self-reported measures of alcohol use in detecting heavy alcohol consumption and AUROC comparisons.**

|  | AUROC (95% CI) | Sensitivity | Specificity | Threshold |
|---|---|---|---|---|
| RSOD (1) | 0.768 (0.706; 0.820) | 75.47% | 70.11% | ≥weekly |
| Twelve-month alcohol use (2) | 0.813 (0.758; 0.863) | 75.47% | 78.74% | >15 |
| Previous-week alcohol use (3) | 0.706 (0.641; 0.763) | 66.04% | 66.09% | ≥19 |
| AUDIT-C (4) | 0.7379 (0.678; 0.796) | 62.26% | 72.41% | ≥8 |
| AUDIT (5) | 0.737 (0.673; 0.792) | 67.92% | 68.39% | ≥15 |

AUROC: Area under the receiver operating characteristics curve; RSOD: Risky single-occasion drinking; AUDIT-C: The Alcohol Use Disorders Identification Test-Consumption; AUDIT: The Alcohol Use Disorders Identification Test.

AUROC comparisons: 1–2: chi2(1) = 3.22, p = 0.073; 1–3: chi2(1) = 2.28, p = 0.131; 1–4*: chi2(1) = 0.94, p = 0.332; 1–5: chi2(1) = 0.95, p = 0.329; 2–3: chi2(1) = 7.53, p = 0.006; 2–4: chi2(1) = 5.12, p = 0.024; 2–5**: chi2(1) = 4.63, p = 0.031; 3–4: chi2(1) = 0.70, p = 0.404; 3–5: chi2(1) = 0.40, p = 0.527; 4–5: chi2(1) = 0.01, p = 0.931.

*This result differs from the result on the sample with hair segment between 3–6 cm: 1–4: chi2(1) = 8.83, p = 0.003.

** This result differs from the result on the sample with hair segment between 3–6 cm: 2–5: chi2(1) = 0.98, p = 0.323.

## Discussion

The main aim of this study was to provide empirical evidence of the psychometric performance of self-reported measures of alcohol use (previous twelve-month alcohol use and previous-week alcohol use), RSOD to assess excessive chronic drinking, as well as the AUDIT and AUDIT-C questionnaires. Empirical evidence was provided by comparing self-reported measures to hEtG, an objective and reliable measure of excessive chronic drinking. We found that the previous twelve-month alcohol use measure had acceptable psychometric performance (sensitivity = 75.5%, specificity = 78.7%) based on a cut-off of >15 drinks per week. RSOD yielded slightly lower performance (sensitivity = 75.5%, specificity = 70.1%) with a cut-off of ≥ weekly. No cut-off gave acceptable psychometric performance for self-reported previous-week alcohol use; even with the best cut-off (≥19), sensitivity and specificity were both lower than 70%. Similarly, the AUDIT and AUDIT-C had lower diagnostic performance to assess chronic excessive drinking.

First, in contrast with our hypotheses, only the previous twelve-month alcohol use measure was found to be a reliable assessment of alcohol excessive chronic drinking among young men. Indeed, although previous-week alcohol use was highly correlated with the previous twelve months of alcohol use, this measure was not associated with acceptable sensitivity and specificity in this study. Considering that the drinking pattern of young men is highly variable from one week to another [29], it was not surprising that the self-reported measure of alcohol use in the previous week had worse psychometric properties than the self-reported measure of alcohol use in the previous twelve months. Indeed, alcohol use in the past week might not be representative of alcohol use during the whole year. These results were consistent with the literature [25, 30]. As proposed by Gmel and his colleagues [25], one may wonder whether higher volumes of alcohol are indicative of a more valid measure or whether they are related to the way alcohol intake was measured. Indeed, the previous-week measure was composed of a diary of the number of drinks consumed in the previous seven days and specified different kinds of alcohol (e.g., beer, white wine, red wine, spirituous), whereas the previous twelve-month measure consisted of a questionnaire separately targeting weekday and weekend consumption over a period of time [25]. Furthermore, the detection window of alcohol use in the hair analysis in our study was between two and six months for at least 70% of the sample.

Second, concerning self-reported RSOD, this study was the first to provide empirical evidence of this measure due to the use of a biomarker. Our results showed that, in addition to

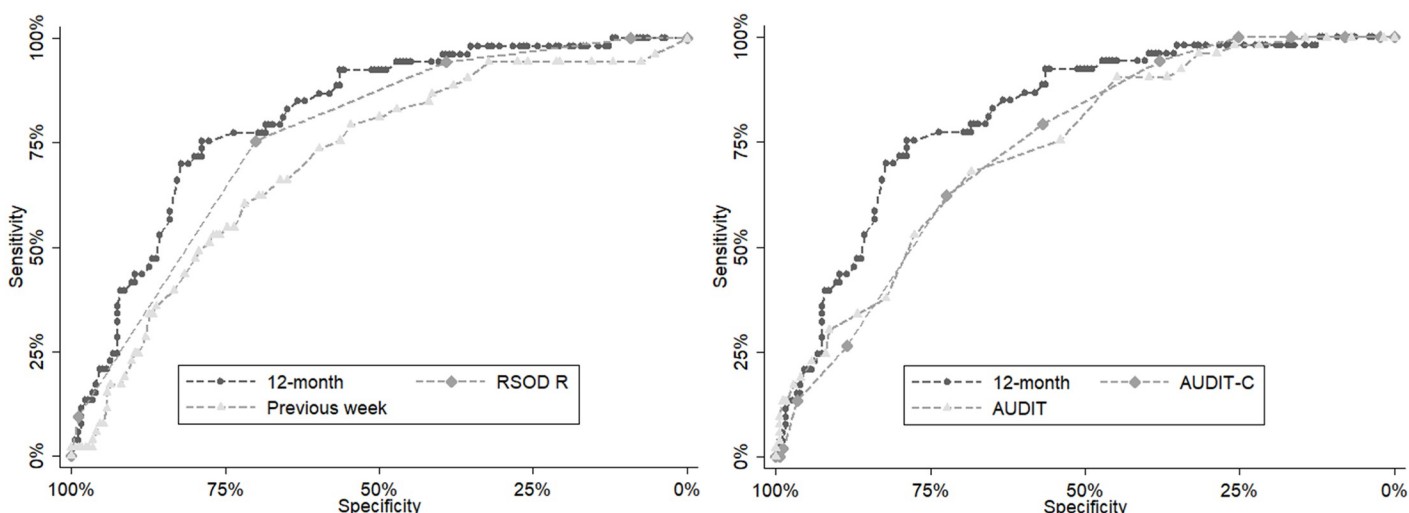

**Fig 1. A comparison of the diagnostic performance of the self-reported measures in detecting heavy alcohol consumption (>60 g/day) using ROC curve analysis.** AUROC, area under the receiver operating characteristics curve. 12-month: Twelve-month alcohol use; RSOD R: Risky single-occasion drinking reversed (from 'daily' to 'less than monthly'); Previous week: Previous-week alcohol use; AUDIT: Alcohol Use Disorder Identification Test; AUDIT-C: The Alcohol Use Disorders Identification Test-Consumption.

acceptable psychometric properties (cut-off ≥ weekly), sensitivity and specificity were maximized for monthly RSOD (sensitivity = 94.3% and specificity = 39.1%) and daily RSOD (sensitivity = 9.4% and specificity = 98.9%). These findings support the usefulness and reliability of the consumption of six alcohol drinks on a single occasion to define an excessive pattern of alcohol use, thereby filling a major gap in this literature. Indeed, an important debate exists about the reliability of this measure (or its US equivalence: the 4/5 measure) to capture risky single-occasion or binge drinking [e.g., 31]. Furthermore, self-reported RSOD was found to be highly associated with self-reported twelve-month alcohol use measures in young men (r = 0.780). Interestingly, although these measures share a large amount of information, after controlling for twelve months of alcohol use, RSOD remained a significant predictor of excessive chronic drinking and still increased the pseudo $R^2$ from 19.9% to 22.3%. These results therefore reinforce the proposal that RSOD may constitute a risk factor for the development of severe alcohol use disorders [e.g., 32, 33] and a reliable indicator for prevention purposes. By

**Table 3. Univariate and multivariate logistic regression assessing to what extent, alcohol use and RSOD were uniquely associated with hEtG.**

| | Univariate | | | | Multivariate | | | | | | | |
|---|---|---|---|---|---|---|---|---|---|---|---|---|
| | OR | 95% CI | p-val | pseudo R2 | OR | 95% CI | p-val | pseudo R2 | | | | |
| RSOD | 7.22 | [3.57; 14.61] | <0.001 | 14.27% | 2.82 | [1.22; 6.52] | 0.016 | 22.29% | | | | |
| Twelve-month alcohol use | 10.65 | [5.18; 21.88] | <0.001 | 19.92% | 6.13 | [2.67; 14.04] | <0.001 | | | | | |
| Previous-week alcohol use | 3.79 | [1.98; 7.25] | <0.001 | 6.96% | n.i. | | | | | | | |
| | Multivariate | | | | Multivariate | | | | Multivariate | | | |
| | OR | 95% CI | p-val | pseudo R2 | OR | 95% CI | p-val | pseudo R2 | OR | 95% CI | p-val | pseudo R2 |
| RSOD | 5.52 | [2.54; 12.03] | <0.001 | 15.20% | n.i. | | | 20.33% | 2.68 | [1.13; 6.34] | 0.025 | 22.38% |
| Twelve-month alcohol use | n.i. | | | | 1.49 | [0.68; 3.25] | <0.001 | | 5.79 | [2.46; 13.61] | <0.001 | |
| Previous-week alcohol use | 1.79 | [0.85; 3.78] | 0.128 | | 8.86 | [4.00; 19.64] | <0.001 | | 1.22 | [0.55; 2.72] | 0.630 | |

n.i.: not included; RSOD: Risky single-occasion drinking; OR: odds ratio; CI: confidence interval; hEtG: ethyl glucuronide in hair.

providing empirical evidence of both RSOD assessment and its association with subsequent excessive chronic drinking, this research offers substantial insights for future studies. Indeed, RSOD evaluation may be easily implemented in large alcohol screening programs, and we support its usefulness to monitor excessive chronic drinking and to implement adapted prevention actions among young populations. Remarkably, results showed that, when focused on the subsample of participants with hairs between 3 and 6 cm (i.e., the previously recommended length), the RSOD measure still have better sensitivity and specificity than the AUDIT-C, a widely-used tool to assess excessive alcohol use among young people [34].

## Limitations

Even if hEtG is considered the most reliable and convenient biomarker to monitor alcohol use with high sensitivity and specificity [1], this measure has some shortcomings. First, there is still a need for standardization of the analytical protocol to reduce between- and within-value variability. Second, we need information about cosmetic hair treatments, use of hair products, regular hygiene habits, and health status, as they could have some impact on the level of hEtG [1], but this information was not available in our study. Third, if hair length samples less than 3 cm or greater than 6 cm are used, the results should be interpreted with caution [15]. This was the case in 40% of our hair samples, but the analyses on the subsample of hair with a length between 3 and 6 cm were similar, and therefore our results seem not to have been affected by this potential problem.

Concerning the self-reported measures, two limitations should be raised. First, assessment periods of self-reported measures should better refer to the detection window of hEtG, that is, three to six months previously, instead of twelve months. Second, self-reported alcohol use in the previous twelve months might underestimate alcohol use, but as shown by the high correlation between hEtG and this measure, rank order was preserved (heavier drinkers according to the EtG reported more self-reported alcohol use than lighter drinkers).

Finally, our sample was only composed of young men, and further studies should be done on women and other age groups. However, it is worth noting that young men between 20 and 39 years old are more likely to be excessive alcohol drinkers [1, 2], and although alcohol use is common across several subpopulations, the heath burden varies across groups [35]; therefore, young men should be deeply investigated. Future studies should also assess excessive alcohol use and use urine EtG to offer complementary results. This would be particularly interesting for previous-week alcohol use measure, which showed poorer sensitivity and sensibility in this study focused on chronic excessive alcohol use.

Beyond these limitations, this study has several implications. Indeed, the current findings have provided empirical validation of self-reported alcohol consumption measures in the previous twelve months and of the assessment of excessive alcohol use (RSOD) based on a comparison with objective hEtG measures. Self-reported measures thus offer reliable indicators of alcohol use for future studies; these measures are notably useful not only for studies aiming to monitor alcohol consumption but also for studies aiming to select a subgroup of alcohol excessive chronic drinkers and investigate explanatory factors or consequences of harmful drinking patterns.

## Conclusions

Excessive chronic drinking is a health burden that young men are more likely to experience worldwide, and as written by the WHO [4], with better awareness, responses at national, regional and global levels will be better. Thus, self-reported measures of the previous twelve months of alcohol use and RSOD are promising indicators at the population level to monitor

alcohol use in general and excessive chronic drinking in specific to help increase this response. These validated indicators will help to monitor young men's alcohol use, inform policymakers and thus develop adequate prevention policies.

## Supporting information

**S1 Fig. A comparison of the diagnostic performance of the self-reported measures in detecting heavy alcohol consumption (>60 g/day) using ROC curve analysis.** AUROC, area under the receiver operating characteristics curve for sample with hair segment between 3–6 cm. 12-month: Twelve-month alcohol use; RSOD R: Risky single-occasion drinking reversed (from 'daily' to 'less than monthly'); AUDIT: Alcohol Use Disorder Identification Test; AUDIT-C: The Alcohol Use Disorders Identification Test-Consumption.
(TIF)

**S1 Table. Self-reported measures of alcohol consumption and hEtG for the whole sample and by level of hEtG for sample with hair segment between 3–6 cm.** [1]: mean +/- sd for quantitative variables and % (n) for categorical variables; sd: standard deviation; p50: median; p25: first quartile; p75: third quartile; RSOD: Risky single-occasion drinking; AUDIT-C: The Alcohol Use Disorders Identification Test-Consumption; AUDIT: Alcohol Use Disorder Identification Test; hEtG: ethyl glucuronide in hair.
(DOCX)

**S2 Table. The diagnostic performance of self-reported measures of alcohol use in detecting heavy alcohol consumption and AUROC comparisons for sample with hair segment between 3–6 cm.** AUROC: Area under the receiver operating characteristics curve; RSOD: Risky single-occasion drinking; AUDIT-C: The Alcohol Use Disorders Identification Test-Consumption; AUDIT: The Alcohol Use Disorders Identification Test. AUROC comparisons: 1–2: chi2(1) = 0.33, p = 0.5671; 1–3: chi2(1) = 3.93, p = 0.048; 1–4: chi2(1) = 8.83, p = 0.003; 1–5: chi2(1) = 0.73, p = 0.393; 2–3: chi2(1) = 4.43, p = 0.035; 2–4: chi2(1) = 8.08, p = 0.005; 2–5: chi2(1) = 0.98, p = 0.323; 3–4: chi2(1) = 0.05, p = 0.825; 3–5: chi2(1) = 0.69, p = 0.405; 4–5: chi2(1) = 2.82, p = 0.093.
(DOCX)

**S3 Table. Univariate and multivariate logistic regression assessing to what extent, alcohol use and RSOD were uniquely associated with EtG for sample with hair segment between 3–6 cm.** n.i.: not included; RSOD: Risky single-occasion drinking; OR: odds ratio; CI: confidence interval.
(DOCX)

**S1 Data.**
(XLSX)

**S2 Data.**
(DOCX)

## Acknowledgments

The authors wish to thank Sabine Jacot for her unwavering support during this study; Maria Sole Maimone, Laura Colombo, and Alexandra Charpentier for the data collection; C-SURF for access to their cohort; and all the participants of C-SURF who participated in this study.

## Author Contributions

**Conceptualization:** Katia Iglesias, Frank Sporkert, Jean-Bernard Daeppen, Gerhard Gmel, Stéphanie Baggio.

**Formal analysis:** Katia Iglesias.

**Funding acquisition:** Katia Iglesias, Stéphanie Baggio.

**Investigation:** Katia Iglesias, Stéphanie Baggio.

**Methodology:** Katia Iglesias, Stéphanie Baggio.

**Writing – original draft:** Katia Iglesias.

**Writing – review & editing:** Séverine Lannoy, Frank Sporkert, Jean-Bernard Daeppen, Gerhard Gmel, Stéphanie Baggio.

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
