## [Decision Letter · Decision Letter 0]

19 Oct 2020

PONE-D-20-26117

Performance of self-reported measures of alcohol use and of harmful drinking patterns against ethyl glucuronide hair testing among young Swiss men

PLOS ONE

Dear Dr. Iglesias,

Thank you for submitting your manuscript to PLOS ONE. After careful consideration, we feel that it has merit but does not fully meet PLOS ONE’s publication criteria as it currently stands. Therefore, we invite you to submit a revised version of the manuscript that addresses the points raised during the review process.

We look forward to receiving your revised manuscript.

Kind regards,

Joel Msafiri Francis, MD, MS, PhD

Academic Editor

PLOS ONE

Journal Requirements:

Reviewers' comments:

Reviewer's Responses to Questions

**Comments to the Author**

1. Is the manuscript technically sound, and do the data support the conclusions?

Reviewer #1: Yes

Reviewer #2: Yes

Reviewer #3: Yes

2. Has the statistical analysis been performed appropriately and rigorously? 

Reviewer #1: Yes

Reviewer #2: Yes

Reviewer #3: Yes

3. Have the authors made all data underlying the findings in their manuscript fully available?

Reviewer #1: Yes

Reviewer #2: Yes

Reviewer #3: No

4. Is the manuscript presented in an intelligible fashion and written in standard English?

Reviewer #1: Yes

Reviewer #2: Yes

Reviewer #3: Yes

5. Review Comments to the Author

Reviewer #1: This is a very interesting and well-written paper that presents results from a study comparing self-reported alcohol measures to the EtG biomarker. Given the limitations of self-report, the paper has high public health significance and the study is well-designed and executed, and clearly described. I have only a few relatively minor comments for the authors to consider.

-Were any other demographic characteristics available among the study sample? It may be helpful to present them. If so, are there any characteristics that may be worth checking as possible moderators? That is, are there subgroups for whom it would be possible that the sensitivity/specificity may differ?

-The authors appropriately describe a limitation of the study being that only young men were included and that future studies should include women and additional age groups. I was also wondering about this specific population of young men who were recruited initially from army recruitment centers. Can the authors comment on the generalizability of this specific population to young men more broadly in Switzerland? Presentation of additional demographic characteristics about the sample may also help in this regard.

-The AUDIT was used in the study but it does not seem like it was used in the sensitivity/specificity ROC analyses. What is the rationale for not using it since it is such a widely used self-report screening tool. The findings of the RSOD indicator being a psychometrically sound indicator were really interesting and possibly adds some evidence to the use of the ‘AUDIT-3’, which refers to screening with only the binge drinking question of the AUDIT (i.e., AUDIT question # 3). It may be useful for the authors to describe some of the literature around the AUDIT-3 in relation to how their RSOD indicator performed here. I believe there have been some studies where the AUDIT-3 did not perform so well.

Reviewer #2: The authors described an interesting statistical approach to evaluate alcohol chronic use, by putting into correlation EtG in hair concentrations and two different self report methods.

The study appears well designed and clearly discussed.

However, main concerns about the article are:

1. The authors did not include the concentrations of EtG ranging from 5 to 30 pg/mg in the evaluation. Since the LOQ of their method would allow to detect and quantify also EtG in the above range, it could be of great interest to include a statistical evaluation of those subjects

2. The authors discussed in details the reliability of EtG in hair in evaluating a chronic excessive alcohol misuse. However, binge drinking is often the most frequent alcohol misuse habit among young people. Hence, limitations on the use of hEtG for such a purpose must be better discussed in the text

3. Though the tables are clear, I would appreciate a further table with only hEtG results (mean, median, min, max, etc.). Maybe it could be added as supplemental material

4. The authors tentatively evaluated the last-week alcohol consumption through the self report measure. Why did they not include a EtG in urine test, together with the hEtG?. Though it is not a marker of chronic alcohol use, it could provide an important additional information on alcohol use during last week. Please make a comment on that issue

Reviewer #3: The paper is well written and organized. The conclusions are supported by the results and a good discussion is provided. However, I have a few additional comments/questions to the authors:

a) Section 2.1. (Biological specimens): Please provide the minimum length of the collected hair samples, or at least the desired length for results interpretation.

b) Section 2.3. (Measures): Please provide the used questionnaire as supplementary material.

c) How were EtG concentrations compared to self-reported data for small hair lengths relatively to the past 12 month’s alcohol use? The authors have disclosed this situation (the detection window of alcohol use in the hair analysis in our study was between two and six months for at least 70% of the sample, in their own words), but perhaps it would be important to go a little bit further in the discussion of this issue.

6. PLOS authors have the option to publish the peer review history of their article (what does this mean?). If published, this will include your full peer review and any attached files.

Reviewer #1: No

Reviewer #2: No

Reviewer #3: No

---

## [Author Response · Author response to Decision Letter 0]

21 Nov 2020

Revision of our Manuscript PONE-D-20-26117 entitled "Performance of self-reported measures of alcohol use and of harmful drinking pattern compared to ethyl glucuronide hair testing among young Swiss men"

Reviewer #1: 

Reviewer: This is a very interesting and well-written paper that presents results from a study comparing self-reported alcohol measures to the EtG biomarker. Given the limitations of self-report, the paper has high public health significance and the study is well-designed and executed, and clearly described. I have only a few relatively minor comments for the authors to consider.

Authors: Thank you your valuable comments and for your positive feedback. Your comments helped us to improve the paper.

Q1.1.

Reviewer: -Were any other demographic characteristics available among the study sample? It may be helpful to present them. If so, are there any characteristics that may be worth checking as possible moderators? That is, are there subgroups for whom it would be possible that the sensitivity/specificity may differ?

Authors: The Reviewer raises an interesting point, but unfortunately, we did not collect other demographic characteristics in our study “SADYSM”. However, as our data were collected from the ongoing Cohort Study on Substance Use and Risk Factors (C-SURF), we have information regarding “Perceived family income” from the initial C-SURF’s questionnaire in 2010, this has been added in the revised manuscript.

Findings showed that 10.1% of our sample did not answer this question in 2010, 41.0% have a perceived family income above the average, 40.1% have a perceived family income in the average, and 8.8% have a perceived family income below the average. For non-excessive chronic drinkers (hEtG < 30), 10.9% did not answer, 35.6% have a perceived family income above the average, 43.7% have a perceived family income in the average, and 9.8% have a perceived family income below the average. For excessive chronic drinkers (hEtG ≥ 30), 7.5% did not answer, 58.5% have a perceived family income above the average, 28.3% have a perceived family income in the average, and 5.7% have a perceived family income below the average. 

Due to the small number of persons with a perceived family income below the average, we combined this response with “perceived family income in the average” response. The diagnostic performance of self-reported measures of alcohol use in detecting heavy alcohol consumption for participants with “perceived family income in or below the average” and for participants with “perceived family income above the average” are similar to the results presented in the Table 2. Importantly, controlling our analyses for perceived family income did not change the results.

It should be noted that the C-SURF constitutes a very large database in which other demographic characteristics may be found (i.e., participants’ higher level of education, parents’ level of education). However, the current authorization we have for this study does not allow us to access to other data such as the level of education (which, in addition, changed over time in this cohort of emerging adults). As the use of perceived family income showed no difference in our results, we hypothesized that socio-economic or educational factors would not impact the sensitivity and specificity results. Therefore, we only included data about perceived family income in the revised manuscript. However, if the Reviewer or the Editor think that such measures are needed, we can make a request to access these data and include them in the paper.

Now the Measures section (p.10, lines 181-184) reads as follow:

“Perceived family income. Data were collected in the C-SURF cohort in 2010-2011 and merged with the SADYSM data. Participants were asked to report their perception of their family income: “How well off is your family compared to other families in your country?”. Participants were grouped in two categories: in the average or below the average vs. above the average.”

Now the Descriptive statistics section (p.12, lines 212-218) reads as follow:

“Participants were, on average, 27.01 � 1.45 years old. From the first C-SURF questionnaire (2010-2011), 41.0% of our sample have a perceived family income above the average, 48.9% have a perceived family income in the average or below (10.1% missing data). For non-excessive chronic drinkers (hEtG < 30), 35.6% have a perceived family income above the average, 53.9% have a perceived family income in the average or below (with 10.9% missing data). For excessive chronic drinkers (hEtG ≥ 30), 58.5% have a perceived family income above the average, 34.0% have a perceived family income in the average or below (with 7.5% missing data).”

Now the ROC analysis section (p.13, lines 233-234) reads as follow:

“The results were similar when taking into account for “perceived family income in or below the average” and “perceived family income above the average”

Q1.2.

Reviewer: -The authors appropriately describe a limitation of the study being that only young men were included and that future studies should include women and additional age groups. I was also wondering about this specific population of young men who were recruited initially from army recruitment centers. Can the authors comment on the generalizability of this specific population to young men more broadly in Switzerland? Presentation of additional demographic characteristics about the sample may also help in this regard. 

Authors: Even if the enrolment phase of C-SURF was done at army recruitment centers in Switzerland, the sample is representative of the population of young Swiss men, as the military recruitment is mandatory in Switzerland. All males around age 20 are evaluated to determine their eligibility for military service, civil service, or no service. There is no pre-selection for this conscription. All conscripts were given a written information sheet and a consent form, as well as a five-minute questionnaire containing questions on demography, alcohol, tobacco, and cannabis use. For further details on the enrolment procedure see [19]. This point was clarified in the manuscript.

Now the Materials and methods section (p.7, lines 118-120) reads as follow:

“Data were collected among young Swiss men from the ongoing Cohort Study on Substance Use and Risk Factors (C-SURF) [18,19]. The C-SURF cohort is composed of young Swiss men initially enrolled in army recruitment centers in 2010 (mean age at that time of enrollment, 20 years old). The military recruitment is mandatory in Switzerland. All males around age 20 are evaluated to determine their eligibility for military service, civil service, or no service without any pre-selection for this conscription. [19]. A subsample of this cohort was invited to participate in a new study (Screening for Alcohol Dependence among Young Swiss Men; SADYSM). SADYSM’s participants were all alcohol users during the previous twelve months.”

Q1.3.

Reviewer: -The AUDIT was used in the study but it does not seem like it was used in the sensitivity/specificity ROC analyses. What is the rationale for not using it since it is such a widely used self-report screening tool. The findings of the RSOD indicator being a psychometrically sound indicator were really interesting and possibly adds some evidence to the use of the ‘AUDIT-3’, which refers to screening with only the binge drinking question of the AUDIT (i.e., AUDIT question # 3). It may be useful for the authors to describe some of the literature around the AUDIT-3 in relation to how their RSOD indicator performed here. I believe there have been some studies where the AUDIT-3 did not perform so well.

Authors: Thank you for the comment, there is actually no specific reason not to use the AUDIT in the sensitivity/specificity analyses. We followed the Reviewer’s suggestion and added the information about both AUDIT-C (i.e., AUDIT-Consumption; the three first items of the AUDIT) and AUDIT in Tables 1 and 2 (as in S1 and S2). Interestingly, even if the AUDIT is a widely used self-report screening tool, its psychometric properties are not as good as expected, both to assess excessive chronic drinking (as shown in this study) and to assess alcohol use disorder (see Baggio and Iglesias discussed in a letter to the Editor in Drug and Alcohol Dependence in 2020 (10.1016/j.drugalcdep.2019.107662). 

Regarding the Reviewer’s suggestion to discuss the results obtained with our RSOD measure and the literature focusing on AUDIT-3, we have expanded the discussion. However, we would like to underline that the aim of this paper was not to perform comparisons between measures nor to evaluate binge drinking or excessive episodic drinking, but to assess chronic excessive drinking. To this end, we can observe that our RSOD measure have better sensitivity and specificity than the AUDIT-C when focusing on the subsample of participants with hairs lengths between 3 and 6 cm (i.e., the recommended length for hEtG measures). 

Please see below the changes regarding the implementation of analyses with the AUDIT.

Now the Measures section (p.10, lines 174-180) reads as follow:

“Alcohol Use Disorder Identification Test. The AUDIT is a ten-item questionnaire to screen for alcohol use disorders [20] and is also used as a screening tool for excessive chronic drinking [26]. The AUDIT includes questions on dependence (3 items), on specific consequences of harmful alcohol use (4 items) and on hazardous alcohol use (3 items). The score ranges between 0 and 40.

AUDIT-C: The Alcohol Use Disorders Identification Test-Consumption. The AUDIT-C includes the three first questions of the AUDIT, which focus on alcohol consumption (frequency, quantity, and excessive drinking episodes) [27; 28]. The score range between 0 and 12.”

Now the Statistical analysis section (p.11, lines 189-208) reads as follow:

“Second, to assess the psychometric properties of self-reported measures of alcohol use and of harmful drinking patterns to assess excessive chronic drinking, sensitivity and specificity were calculated for the optimal cut-off values selected from the receiver operating characteristic (ROC) curves. The ROC curves of self-reported measures were compared using the areas under the ROC curves (AUROCs) to assess the best self-reported measure of excessive chronic drinking. The AUDIT and AUDIT-C tools were also tested, as they are widely used as self-reported screening tools. […]. Finally, as the SoHT [15] recommended hair length between 3 and 6 cm, a sensitivity analysis was run on a subsample of 129 participants with hair length between 3 and 6 cm. The overall results remained the same in this subsample, only two exceptions were noted in the comparisons between self-reported measures (i.e., between RSOD and AUDIT-C AUROC and AUDIT-C and Twelve-month alcohol use AUROC; see Table 2). Therefore, only the results for the whole sample are presented in the paper and the unique difference was mentioned in the results. The analyses of the 129 participants are available in the supplementary material (S1 Fig, S1-S3 Tables).”

Now the ROC analysis section (p.13, lines 227-232) reads as follow:

“Twelve-month alcohol use had the best sensitivity and specificity to assess excessive chronic drinking with a cut-off of 15 drinks per week (75.5% and 78.7%, respectively), which were significantly better than the findings regarding previous-week alcohol use (p=0.006), AUDIT-C (p=0.024), and AUDIT (p=0.314), but not significantly better than the findings regarding RSOD (p=0.071) when AUROCs were compared. For the hair segment between 3-6 cm, RSOD was significantly better than AUDIT-C (p = 0.003) when AUROCs were compared. ”

Tables 1-2, and Fig.1 modified

Now the Discussion section (p.14, line 259 and p.16, line 308) reads as follow:

“The main aim of this study was to provide empirical evidence of the psychometric performance of self-reported measures of alcohol use (previous twelve-month alcohol use and previous-week alcohol use), RSOD to assess excessive chronic drinking, as well as the AUDIT and AUDIT-C questionnaires. Empirical evidence was provided by comparing self-reported measures to hEtG, an objective and reliable measure of excessive chronic drinking. We found that the previous twelve-month alcohol use measure had acceptable psychometric performance (sensitivity=75.5%, specificity=78.7%) based on a cut-off of >15 drinks per week. RSOD yielded slightly lower performance (sensitivity=75.5%, specificity=70.1%) with a cut-off of ≥ weekly. No cut-off gave acceptable psychometric performance for self-reported previous-week alcohol use; even with the best cut-off (≥19), sensitivity and specificity were both lower than 70%. Similarly, the AUDIT and AUDIT-C had lower diagnostic performance to assess chronic excessive drinking.” […] “Remarkably, results showed that, when focused on the subsample of participants with hairs between 3 and 6 cm (i.e., the previously recommended length), the RSOD measure still have better sensitivity and specificity than the AUDIT-C, a widely-used tool to assess excessive alcohol use among young people [34].”

We added these three references: 

27. Bush, K.. The AUDIT alcohol consumption questions (AUDIT-C). An effective brief screening test for problem drinking. Archives of Internal Medicine. 1998, 158(16), 1789. 

28. Tuunanen, M,Aalto M, Seppä K. Binge drinking and its detection among middle-aged men using AUDIT, AUDIT-C and AUDIT-3. Drug Alcohol Rev. 2007, 26:295–9.

34. Liskola, J., Haravuori, H., Lindberg, N., Niemel., S., Karlsson, L., Kiviruusu, O., et al. AUDIT and AUDIT-C as screening instruments for alcohol problem use in adolescents. Drug and Alcohol Dependence, 2018, 188, 266–273.

And in the Supporting information material, S1-S2 Tables, and S1 Fig. modified.

Reviewer #2: 

Reviewer: The authors described an interesting statistical approach to evaluate alcohol chronic use, by putting into correlation EtG in hair concentrations and two different self report methods. The study appears well designed and clearly discussed.

Authors: Thank you very much for reviewing this paper and for your positive feedback. Your comments helped us to improve the paper

Q2.1.

Reviewer: -However, main concerns about the article are: 1. The authors did not include the concentrations of EtG ranging from 5 to 30 pg/mg in the evaluation. Since the LOQ of their method would allow to detect and quantify also EtG in the above range, it could be of great interest to include a statistical evaluation of those subjects

Authors: Thank you for the comment. We probably were not clear enough regarding this point, as we actually took all the concentrations of hEtG ranging from 0 to 691 (the maximum value found in our sample), i.e., with hEtG lower than 0 being defined as non-excessive chronic drinking and hEtG equal or greater than 30 as excessive drinking. This information was added to the Measure section, that reads now as follow (page 9, lines 159-161):

“Excessive chronic drinking. Excessive chronic drinking was based on the threshold of 30 pg/mg, with hEtG < 30 being non-excessive chronic drinking and hEtG ≥ 30 being excessive chronic drinking.”

Q2.2.

Reviewer: -2. The authors discussed in details the reliability of EtG in hair in evaluating a chronic excessive alcohol misuse. However, binge drinking is often the most frequent alcohol misuse habit among young people. Hence, limitations on the use of hEtG for such a purpose must be better discussed in the text

Authors: We completely agree with the fact that hEtG has some limitations as a gold standard of excessive drinking, as already mentioned in the limitations of the first version of our paper. We also acknowledge that binge drinking is a frequent alcohol habit among young people. However, the purpose of the paper was to go beyond binge drinking and assess the diagnostic performance of the most used self-reported measures of alcohol use. While the focus on binge drinking is certainly interesting, it was not the main aim of our study and we believe that the alcohol literature currently needs evidence regarding the reliability of self-reported measures in general. Indeed, it is important to be able to properly capture chronic excessive alcohol use to target participants who need alcohol-related preventions and interventions.

Q2.3.

Reviewer: -3. Though the tables are clear, I would appreciate a further table with only hEtG results (mean, median, min, max, etc.). Maybe it could be added as supplemental material

Authors: Please see these descriptive statistics in Table 1 for the whole sample (n=227), for the participants having an hEtG lower than 30, and for those having an hEtG equal or greater than 30 and in S1 Table for the sample with hair segment between 3-6 cm (n=129).

Q2.4.

Reviewer: -4. The authors tentatively evaluated the last-week alcohol consumption through the self report measure. Why did they not include a EtG in urine test, together with the hEtG?. Though it is not a marker of chronic alcohol use, it could provide an important additional information on alcohol use during last week. Please make a comment on that issue

Authors: Thank you for this suggestion, we agree that it would have been very interesting to have EtG in urine to better inform about past-week alcohol use. However, as this study specifically focused on chronic excessive drinking, we did not collect urine samples. Following the Reviewer’s suggestion, we now mention in the discussion that the measure of past-week alcohol consumption has the poorer specificity and sensitivity to assess chronic excessive drinking and that future studies should use urine measures to offer complementary information on the reliability of this measure to assess excessive drinking. 

Now the Limitations section (p.17, line 325 and p. 18, line 332) reads as follow:

“Finally, our sample was only composed of young men, and further studies should be done on women and other age groups. However, it is worth noting that young men between 20 and 39 years old are more likely to be excessive alcohol drinkers [1,2], and although alcohol use is common across several subpopulations, the heath burden varies across groups [35]; therefore, young men should be deeply investigated. Future studies should also assess excessive alcohol use and use urine EtG to offer complementary results. This would be particularly interesting for previous-week alcohol use measure, which showed poorer sensitivity and sensibility in this study focused on chronic excessive alcohol use.”

Reviewer #3: 

The paper is well written and organized. The conclusions are supported by the results and a good discussion is provided. However, I have a few additional comments/questions to the authors: 

Authors: Thank you very much for reviewing this paper and for your positive feedback. Your comments helped us to improve the paper

Q3.1.

Reviewer: -a) Section 2.1. (Biological specimens): Please provide the minimum length of the collected hair samples, or at least the desired length for results interpretation. 

Authors: This information has been moved at the end of the Biological specimens section for better clarity (p.8, lines 129-133), we thank the Reviewer for the comment.

“Among the 233 young men recruited for the study, 227 agreed to give a hair sample for ethyl glucuronide determination. The length of the hair strands varied from 0.5 to 25 cm (mean: 4.2 cm, median: 3.0 cm). The assessments also included self-reported measures of alcohol use (see measures section). Data were collected at the Lausanne University Hospital (Switzerland) from October 2017 to June 2018.”

Q3.2.

Reviewer: -b) Section 2.3. (Measures): Please provide the used questionnaire as supplementary material. 

Authors: We provided the questionnaire as supplementary material as suggested.

Q3.3.

Reviewer: -c) How were EtG concentrations compared to self-reported data for small hair lengths relatively to the past 12 month’s alcohol use? The authors have disclosed this situation (the detection window of alcohol use in the hair analysis in our study was between two and six months for at least 70% of the sample, in their own words), but perhaps it would be important to go a little bit further in the discussion of this issue.

Authors: The SoHT recommended hair length between 3 and 6 cm, and as mentioned in the limitations the paper, results from hair length samples less than 3 cm should be interpreted with caution. As suggested by the reviewer, we investigated a little bit more the hair length samples less than 3 cm. Interestingly and as expected due to the results of the whole sample and on the “adequate sample”, the results of the hair length samples less than 3 cm, remain the same for 12 month’s alcohol use (see table in the response to reviewers).

We also investigated the distribution of hEtG and of 12 month’s alcohol use per hair length samples (see tables in the response to reviewers).

Therefore, we mention that our findings remained the same with different lengths of hairs. We already included a sensitivity analysis in the manuscript. This second sensitivity analysis confirmed our findings and seemed to indicate that the chronic excessive drinking was constant over time for this sample. You can find the description of the sensitivity analysis in the method section, which reads as follow (p.11, lines 201-208):

“Finally, as the SoHT [15] recommended hair length between 3 and 6 cm, a sensitivity analysis was run on a subsample of 129 participants with hair length between 3 and 6 cm. The overall results remained the same in this subsample, only two exceptions were noted in the comparisons between self-reported measures (i.e., between RSOD and AUDIT-C AUROC and AUDIT-C and Twelve-month alcohol use AUROC; see Table 2). Therefore, only the results for the whole sample are presented in the paper and the unique difference was mentioned in the results. The analyses of the 129 participants are available in the supplementary material (S1 Fig, S1-S3 Tables).”

---

## [Decision Letter · Decision Letter 1]

8 Dec 2020

Performance of self-reported measures of alcohol use and of harmful drinking patterns against ethyl glucuronide hair testing among young Swiss men

PONE-D-20-26117R1

Dear Dr. Iglesias,

We’re pleased to inform you that your manuscript has been judged scientifically suitable for publication and will be formally accepted for publication once it meets all outstanding technical requirements.

Kind regards,

Joel Msafiri Francis, MD, MS, PhD

Academic Editor

PLOS ONE

Additional Editor Comments (optional):

Reviewers' comments:

Reviewer's Responses to Questions

**Comments to the Author**

1. If the authors have adequately addressed your comments raised in a previous round of review and you feel that this manuscript is now acceptable for publication, you may indicate that here to bypass the “Comments to the Author” section, enter your conflict of interest statement in the “Confidential to Editor” section, and submit your "Accept" recommendation.

Reviewer #2: All comments have been addressed

Reviewer #3: All comments have been addressed

2. Is the manuscript technically sound, and do the data support the conclusions?

Reviewer #2: Yes

Reviewer #3: Yes

3. Has the statistical analysis been performed appropriately and rigorously? 

Reviewer #2: Yes

Reviewer #3: Yes

4. Have the authors made all data underlying the findings in their manuscript fully available?

Reviewer #2: Yes

Reviewer #3: No

5. Is the manuscript presented in an intelligible fashion and written in standard English?

Reviewer #2: Yes

Reviewer #3: Yes

6. Review Comments to the Author

Reviewer #2: (No Response)

Reviewer #3: (No Response)

7. PLOS authors have the option to publish the peer review history of their article (what does this mean?). If published, this will include your full peer review and any attached files.

Reviewer #2: No

Reviewer #3: **Yes: **Mário Barroso

---

## [Editor Report · Acceptance letter]

10 Dec 2020

PONE-D-20-26117R1 

Performance of self-reported measures of alcohol use and of harmful drinking patterns against ethyl glucuronide hair testing among young Swiss men 

Dear Dr. Iglesias:

I'm pleased to inform you that your manuscript has been deemed suitable for publication in PLOS ONE. Congratulations! Your manuscript is now with our production department. 

Kind regards, 

on behalf of

Dr. Joel Msafiri Francis 

Academic Editor

PLOS ONE